# Formulation of a High-Quality Cold-Pressed Vegetable Oil (Virgin) Based on a Blend of Four Oilseeds

**DOI:** 10.3390/foods13213347

**Published:** 2024-10-22

**Authors:** Ibtissame Guirrou, Abderraouf El Antari, Abdelhay El Harrak, Abdelghani Nabloussi

**Affiliations:** 1Regional Agricultural Research Center of Meknes, National Institute of Agricultural Research, Avenue Ennasr, P.O. Box 415, Rabat 10090, Morocco; ibtissame.guirrou@inra.ma; 2Laboratory of Biotechnology and Valorization of Bio-Resources (BioVar), Faculty of Science, University Moulay Ismail, P.O. Box 11201, Zitoune, Meknes 50100, Morocco; a.elharrak@umi.ac.ma; 3Regional Agricultural Research Center of Marrakech, National Institute of Agricultural Research, Avenue Ennasr, P.O. Box 415, Rabat 10090, Morocco; abderraouf.elantari@inra.ma

**Keywords:** vegetable oil, formulated blend oil, nutritional profile, antioxidant properties

## Abstract

Vegetable oils are crucial for the human diet, providing energy and essential fatty acids. This study investigates the formulation of a high-quality cold-pressed vegetable oil blend from rapeseed, sunflower, sesame, and safflower, chosen for their agronomic benefits, cost-effectiveness, and reduced environmental impact. For the first time, this study is carried out in order to enhance the nutritional profile of these blend oils compared to commercial oils. The study’s results showed that all formulated blend oils had higher total polyphenol and flavonoid content. Specifically, the blend of 1/2 rapeseed, 1/4 sunflower, 1/8 sesame, and 1/8 safflower had an oil yield ranging from 37 to 39% and was rich in total polyphenols (18 mg GAE/100 g), total flavonoids (2 mg/g), antioxidant activities (52%), oleic acid (46.4%), and saturated fatty acids (11%), with a balanced omega-6/omega-3 ratio (2.5). Consuming this blend oil offers a healthier choice rich in nutrients and natural antioxidants. This could open new market opportunities and cater to the growing demand for healthier oil options, especially since it is extracted without a refining process. Further research could focus on the sensory attributes and consumer acceptance of these blend oils to ensure market success, noting that sesame and sunflower involve agreeable pronounced aromas.

## 1. Introduction

Vegetable oils are major components of the human diet. Several oilseeds are commonly used for the extraction of these vegetable oils. Depending on their nature and processing (virgin or refined oils) and their fatty acid composition profiles, vegetable oils have different nutritional and organoleptic qualities.

Vegetable oils represent dense macronutrients that provide a concentrated source of energy to the body [1]. They are also a source of essential fatty acids (omega-3 and omega-6) that the human body cannot synthesize. Additionally, vegetable oils derived from seeds such as sunflower, rapeseed, and soybeans serve as rich products containing fat-soluble vitamins, including A, D, E, and K. Moreover, these oils are abundant in beneficial compounds such as tocopherols, polyphenols, and coenzyme Q. Besides contributing to taste and color, all of the above micronutrients help protect against health disorders such as cardiovascular diseases and cancer [2,3,4,5]. In recent years, it has been reported that phenolic acids and flavonoids have multifunctional properties and beneficial health effects [6,7]. In particular, polyphenols have attracted a great deal of attention [8,9,10,11]. Oilseeds contain a number of phenolic compounds that contribute to the oxidative stability of the oil and can serve as antioxidants to reduce the stress of oxidation on human health [4,9].

The continuous growth of the human population leads to a high demand for food, which has led to an intensification of agricultural practices for maximum food production [12,13]. In the context of consumer demand for diversified production, new market opportunities for producers, and the globalization of oilseed markets, oilseed industry players (industrials) have developed specialized productions to add value to this sector. Among these special products are erucic rapeseed oil (an ancestral rapeseed rich in long-chain fatty acids, especially erucic acid); oleic oils (rich in oleic acid) or linoleic oils (standard linoleic type) for recognized nutritional and functional qualities; organic oils (Bio), which remain a young market in development and structuring; and, finally, the emergence of a new trend involving the use of blend oils/combined oils by the food industry [14,15,16].

Blend oils/combined oils are available in the markets either pre-mixed for direct consumer use or indirectly by the food industry through incorporation into other food products or for specialized applications. For example, they blend soybean oil with sunflower oil for cooking purposes. The use of blend oils by the food industry comes mainly to reduce the cost of purchasing raw materials while optimizing technical properties or even providing additional functionalities [17]. Market leaders opt to purchase oils separately and create their own blends to optimize fatty acid composition, technical properties, and prices, knowing that the use of oleic oils is more expensive [14]. In contrast, offering an oil obtained directly from a blend of several seeds effectively addresses the nutritional deficiencies of monovarietal oil from a single seed. However, to the best of our knowledge, no scientific work has focused on the characterization of the nutritional value of blended oils obtained from the extraction of a mixture of seeds rather than a mixture of oils. Only a few blended oils on the market specify the types of seeds used, without providing any further details.

From a scientific perspective, recent publications have expanded the understanding of blended vegetable oils beyond studies focused solely on single-seed oil nutritional quality [2,4,14,18,19,20]. Recent studies now include research on the enhancement of oxidative stability through the blending of sunflower oil with moringa and sesame oils [21] and the improvement in soybean oil stability by adding sesame and almond oils [22]. Additionally, a comprehensive review by [23] highlights the benefits of blending oils to achieve a balanced fatty acid profile, improved physicochemical characteristics, and augmented storage stability, emphasizing the growing recognition of oil blending as a method to enhance both the nutritional and functional qualities of vegetable oils. Other authors have reported an approach that relies on the fortification of monovarietal vegetable oils, derived from a single seed, especially with vitamins A, D, and E. However, one of the main challenges encountered in oil fortification is the ability of industries to implement a fortification plan that specifies the type and quantity of micronutrients used, while considering targeted consumers (socio-economic groups and age). This plan must also comply with food safety precautions and the conditions required during production, transportation, storage, and sale. On the other hand, despite the availability of subsidized vitamin premixes, oil fortification remains optional in many parts of the world [1,3,24,25,26].

The development of high-quality vegetable oils from multiple seeds offers a blend of benefits and challenges, influenced by the choice of seeds and extraction methods. Several studies have demonstrated that the qualitative and biochemical characteristics of vegetable oils are generally affected by environmental factors (climate and soil), cultural practices, genotypes, extraction conditions, and storage conditions [5,27,28,29,30,31,32]. This study focuses on creating a virgin vegetable oil derived from a specific mixture of oilseeds. Virgin oils, known for their superior nutritional value compared to refined oils, are extracted through pressing and retain more nutrients and antioxidants. Research indicates significant nutrient loss in refined oils, with reductions of up to 98.6% in carotenoids, 8.5% in tocopherols, 19.5% in phytosterols, and 45% in squalene [32,33]. The popularity of virgin oils is increasing in Europe, leading to the emergence of small and medium-sized industries in countries like Switzerland, Austria, and Germany [27]. Additionally, the UK market for cold-pressed, unrefined rapeseed oil has grown by 34% annually since 2013 [34]. Formulating vegetable oils from multiple seeds, particularly in their virgin state, can optimize flavor, nutritional content, stability, yield, and cost while minimizing environmental impact. The objective of this study was the formulation of a cold-pressed vegetable blend oil derived from four oilseeds, showing the advantages of this formulation in terms of its nutritional profile, compared to commercial oils, which helps to satisfy the growing consumer demand for healthier oil options.

## 2. Materials and Methods

### 2.1. Seeds Used in the Production of This Formulation

The choice of rapeseed, safflower, sunflower, and sesame as seeds for producing this blend oil stems from agronomic and nutritional considerations. From an agronomic point of view, safflower is a species of potential interest for agriculture and industry due to its great adaptability to soil and climatic conditions and its tolerance to water stress, making it the best crop option in arid and semi-arid regions [31,35]. Rapeseed (canola or ‘00’ type) and oil sunflower (standard linoleic type) are high-yielding crops that can be grown in a wide range of soils and climatic conditions; they are good rotation crops, helping to break disease and pest cycles, improving soil health, and allowing better water and nutrient absorption while reducing soil erosion [13,36]. Sesame is also a valuable crop, mainly used for oil production. It is relatively drought-tolerant and can grow in a range of soil types. Moreover, sesame has the advantage of being grown as a catch crop, sown in late spring or early summer, just after cereal harvest [37]. With regard to nutritional importance, the nutritional value of safflower oil is similar to olive oil, being rich in linoleic acid [29,38]. Sesame oil also ranks second, after olive oil, in terms of nutritional value [37]. Rapeseed oil is low in saturated fatty acids and rich in monounsaturated and polyunsaturated fatty acids, which are beneficial for heart health. It is also rich in omega-3 fatty acids and a good source of vitamins and minerals, including vitamin E, potassium, and magnesium [34]. Finally, sunflower oil is known for its high antioxidant activity, allowing better stability and being the most used for cooking [4].

The varieties used are ‘Moufida’ for rapeseed (canola or ‘00’ type), ‘Ichraq’ for sunflower (standard linoleic type), the ‘CM50’ variety for safflower (standard linoleic type), and the ‘ML13’ sesame cultivar (oleic-linoleic standard type), commonly grown in the Tadla region (Beni Mellal-Khenifra Region). These varieties have the standard fatty acid composition of their respective species.

The cultivation of safflower, rapeseed, and sunflower took place at the Experimental INRA Domain of Douyet over two consecutive years (2021 and 2022). This site is situated approximately 10 km from Fes, at coordinates 34°04′ North and 5°07′ West, with an elevation of 416 m above sea level. The area’s soil is classified as vertisol, composed of silt, clay, and sand, respectively, 58%, 29%, and 12%. During the two-year study period (2021–2022), the average annual precipitation was recorded as 317.8 mm and 451.4 mm, respectively. Prior to planting, the soil was enriched with a base fertilizer mixture consisting of 60 units of nitrogen, 80 units of potassium, and 80 units of phosphorus. Later, at the onset of the flowering phase, 40 additional units of nitrogen were applied as a top dressing. For sesame, the soil and climatic conditions were those of the Experimental INRA Domain of Afourare, located in the province of Beni-Mellal (33°55′59″ North, 5°16′28″ West) at an altitude of 446 m. The soil was a chronic luvisol. The average rainfall for the two years of experimentation, 2021 and 2022, was 53 and 185 mm, respectively. However, sesame is a catch crop (planted after cereal harvest) whose water needs, estimated at about 500 mm, are mainly met by irrigation.

### 2.2. Composition of the Formulated Blends Oil and the Moroccan Commercial Oils

Thirteen formulated blends based on rapeseed, sunflower, sesame, and safflower seeds (named no 1 through no 13) and seven samples of commercial oil of different compositions were characterized for oil yield, acid index, peroxide index, iodine index, total polyphenol content, flavonoid content, antioxidant activity, and fatty acid composition, including the omega-6/omega-3 ratio. The seed proportions used for the formulated blends and the Moroccan commercial oils are provided in Table 1.

### 2.3. Extraction Conditions

After weighing and mixing the seeds of the four species according to the previously defined proportions, the seeds were ground. The ground material was then placed directly in the mechanical extractor, with no external temperature application (cold pressing), using the YILMAZ Model/Series 6205ZZ (MES Elektromekanik Döküm, Tekirdağ, Turkey). However, due to friction, the temperature did not exceed 40 °C. It is important to indicate that the critical stage of the virgin oil production chain is the period from seed harvest to processing, as extraction and purification have only a minor influence on the microbiological oil’s quality. In our case, the seeds used were stored before grinding and oil extraction in a cold room at +4 °C and humidity below 40%. The oil thus obtained was filtered. The filtered oil was finally stored in opaque glass bottles before biochemical analysis.

### 2.4. Methods Used for Biochemical Analysis

#### 2.4.1. Oil Yield

Oil was extracted using a mechanical extractor (cold pressing). Oil yield, expressed in %, was calculated using the formula:Oil yield (%) = [M2/M1] × 100(1)
where M1 is the used weight of the ground seeds measured in grams and M2 is the weight of the obtained oil, also in grams.

#### 2.4.2. Acid Index

This index, expressed in mg KOH/g, was determined by titration of a solution of oil in ethanol, in accordance with the method NF EN ISO 660, as updated in September 2009.

#### 2.4.3. Peroxide Index

This index was carried out based on the method NF ISO 3960 from March 2004. Its unit is meq O_2_/kg.

#### 2.4.4. Iodine Index

This index was determined using the WIJ’s solution and by titrating the excess iodine with Na_2_S_2_O_3_ according to the method ISO 3961 from 2018. Its unit is g I_2_/100g.

#### 2.4.5. Extraction of the Phenolic Fraction

Before assessing the content of phenolic, flavonoid, and antioxidant compounds, an extract preparation was performed following the method described by Tsimidou et al. [39]. An oil sample of 2 g was dissolved in a mixture of 10 mL of hexane and 4 mL of 60% methanol (*v*/*v*). After agitation for 2 h at ambient temperature in darkness, the supernatant was filtered using filter paper. Each extraction was conducted in three separate instances. The resulting three filtrates were then merged, cleansed with hexane (10 mL), and preserved at 4 °C pending the following analyses.

#### 2.4.6. Total Polyphenol Content

This content, expressed in mg GAE/100 g, was determined following a modified version of the Folin–Ciocalteu colorimetric technique, adapting the approach described by Singleton et al. [40]. In brief, the oil extract (50 μL) was combined with distilled water (3 mL), Folin–Ciocalteu reagent (250 μL), and sodium carbonate at 7% (750 μL). At ambient temperature, the obtained mixture was stirred for 8 min, before adding distilled water (950 μL). After incubation in darkness for 2 h, the absorbance was measured at 765 nm, with a blank as the reference. The absorbance was measured using a UV–Vis spectrophotometer (Model V-530, Jasco, Germany). For the calibration curve, Gallic acid served as the standard.

#### 2.4.7. Total Flavonoid Content

This analysis was carried out according to Favati et al. [41]. The oil extract (0.5 mL) was combined with aluminum chloride at 2% in a methanol solution (0.5 mL). The obtained mixture was stirred for 15 min at ambient temperature. Then, the absorbance was measured at 430 nm, with methanol serving as the blank. The unit of this parameter is mg/g.

#### 2.4.8. Antioxidant Activity

Expressed in %, this parameter was determined using the DPPH (2,2-diphenyl 1-picrylhydrazyl) technique according to Brand-Williams et al. [42] with slight modifications. Briefly, an aliquot of the oil extract (50 μL) was mixed with DPPH solution (950 μL) prepared in methanol (0.030 mg/mL). The mixture was incubated in darkness for 60 min, after which its absorbance was measured at 515 nm, using ultrapure water as the blank. The percentage of antioxidant activity was calculated using the formula:Antioxidant activity (%) = (Ac − Ae/Ac) × 100(2)
where Ac represents the absorbance of the control, while Ae is the absorbance of the sample.

#### 2.4.9. Fatty Acid Composition

The composition of fatty acids was analyzed using a method based on gas chromatography, in accordance with the guidelines set forth in ISO standard 12966-2 (2017). The initial step involved the transformation of FAs into their methyl ester derivatives (FAMEs). This was achieved by combining the oil (60 mg) with hexane (3 mL) and methanolic potassium hydroxide at 2 N (0.3 mL). The obtained mixture underwent continuous agitation for a period of 25 min to ensure a complete reaction.

The chromatographic analysis was conducted using a Varian CP 3380 system (Varian, Palo Alto, CA, USA). This apparatus was fitted with a CP-Wax 52 CB capillary column, characterized by dimensions of 25 m in length, a 0.25 mm internal diameter, and a 0.20 μm film thickness. The system configuration included a split–splitless injection port, an automated sample introduction system (CP-8400), and detection via flame ionization (FID). The analytical parameters were optimized as follows: the sample was introduced at 220 °C, the detector was maintained at 230 °C, and the column oven was held isothermally at 190 °C. High-purity nitrogen served as the mobile phase.

For each analysis, a sample volume of 1 μL was introduced into the system with a split ratio of 1:50. To ensure statistical reliability, each sample was analyzed in triplicate. The quantitative results were expressed as percentages of individual fatty acids relative to the total FA content, calculated by normalizing the areas of the chromatographic signals. To facilitate accurate identification and quantification, a commercially available mixture of FAME standards (encompassing chain lengths from C4 to C24, designated as FAME Mix 37) was employed. The identification of individual components was based on the comparison of retention times with these known standards.

### 2.5. Statistical Analysis

A one-way analysis of variance (ANOVA 1) was performed to assess the variability and its significance among all formulated blend oils and the Moroccan commercial oils based on biochemical data. The mean values were compared using Duncan’s test at a 5% significance level. Statistical analysis was carried out using SPSS software version 28 (IBM, Armonk, NY, USA).

## 3. Results

### 3.1. Variability Among the Formulated Oils and the Commercial Ones

Table 2 presents the results of the ANOVA analysis for all studied oils, including 17 formulated oils (13 blended oils and their 4 single oils) and 7 Moroccan commercial oils, analyzed separately. The statistical results demonstrated that all studied oils, regardless of their category (formulated or commercial), exhibited highly significant differences (*p* < 0.001) across the investigated parameters: oil yield, acid index, peroxide index, iodine index, polyphenol content, flavonoid content, antioxidant activity, and fatty acid composition, including the omega-6/omega-3 ratio. This indicates remarkable variation and distinction among these oils. Furthermore, by analyzing the contrast between the formulated oils and the commercial ones, significant differences were observed between these two groups for all the parameters except for antioxidant activity. The oil yield was excluded from this contrast analysis, as we did not have any information on this parameter for commercial oils.

Regarding the obtained results of the acid, peroxide, and iodine indexes, all the studied oils—whether formulated by us or available on the market—met the standards of the Codex Alimentarius (Standard for Named Vegetable Oils—CXS 210-1999, amended in 2023), which normalize the quality characteristics of vegetable oils. The acid index values were less than 4 mg KOH/g oil, as recommended for virgin oils. The peroxide index values were below 15 milliequivalent of active oxygen/kg oil, which is the maximum recommended level for virgin oils. The peroxide index can be viewed as an indicator of the oxidative state, while the acidity index reveals some information about the degree of lipid unsaturation, which is related to the rancidity of the oil [43,44]. Therefore, the quality of all the oils herein studied was deemed satisfactory.

### 3.2. Oil Yield

According to the industrial side, the oil yield of oilseeds is a very important criterion. The choice of rapeseed, safflower, sunflower, and sesame also responds to this criterion, as sesame has the highest oil yield, ranging from 40% to 63% [45], followed by rapeseed (38% to 44%) [5], sunflower (36% to 42%) [46], and finally safflower (25% to 35%) [35].

As confirmed by our study, oil yield varies significantly across different oil sources, with safflower having the lowest mean oil yield (25.96%) and sesame having the highest (49.36%) (Figure 1). Rapeseed and sunflower oil samples, along with formulated blends no 1, no 9, no 2, no 3, no 11, and no 12, have intermediate content values, ranging from 39.35% to 43.58%. Then, we found the other formulated blends oils no 7, no 4, no 10, no 13, and no 8 to have oil yields ranging from 36.18% to 38.77%. Knowing that the most consumed oilseed in the world is soybean [16,22,47], all these formulated blend oils are much richer in oil than this (17% to 20%) [48].

### 3.3. Total Polyphenol and Flavonoids Content

The results of total polyphenol content (TPC) and flavonoid content (TFC) are presented in Figure 2. It can be observed that all formulated blend oils are richer in TPC and TFC compared to commercial oils. The formulated blend oils, especially no 7, no 6, no 8, no 4, and no 9, have exceptionally high TPC values (18–20 mg GAE/100 g), indicating superior nutritional quality. Additionally, the formulated blend oils no 12, no 10, no 13, and no 2 also show competitive TPC values, often outperforming or matching the TPC of single oils extracted from sesame and sunflower (11–15 mg GAE/100 g). Following these, the formulated blends no 11, no 3, no 1, and no 5 are grouped together, indicating a similar classification to single oils of safflower and rapeseed (9–11 mg GAE/100 g).

Finally, we found that the commercial oils presented TPC values ranging from 5 to 9 mg GAE/100 g. Compared to soybean oil (1–4 mg GAE/100 g), known as the most consumed in the world, and olive oil (15–50 mg GAE/100 g), recognized as a high-quality nutritional oil by consumers [49,50,51,52], the first class of formulated blend oils represents a good source of oil with a rich total polyphenol content.

Similarly, the study demonstrates that some formulated blend oils are rich in TFC. Notably, the formulated blend oils no 12, no 13, no 8, and no 7 have exceptionally high TFC values, ranging from 1.91 to 2.41 mg/g. Following these blends, oils no 11, no 10, and no 9, as well as single oils of safflower and sesame, exhibit TFC values between 1.57 and 1.83 mg/g. However, commercial oils such as the blend of sunflower/rapeseed, the blend of sunflower/soybean, and the single oils of sunflower, soybean, and rapeseed exhibit the lowest TFC values, ranging from 0.49 to 0.74 mg/g. It was also reported that TFC ranged from 0.32 to 0.65 mg/g in soybean oil [49,53] and from 0.13 to 1.7 mg/g in olive oil [54,55,56].

The total phenolic and flavonoid compounds, in addition to antioxidant activity, provide information on the degree of the oil’s resistance to oxidation [4,9]. The study of these parameters can successfully guide the selection of better oils and oil mixtures. Our findings highlight the potential of using these proposed formulated blends, with elevated TPC and TFC, in developing high-quality vegetable oils.

### 3.4. Antioxidant Activity

In this study, the results of the antioxidant activity investigated for the formulated blend oils and commercial oils indicate that formulated blend oils exhibit slightly lower antioxidant activity compared to their commercial counterparts (Figure 3). This difference can be attributed to the presence of food additives such as vitamins E and A in commercial oils, which enhance their shelf life, oxidation resistance, and stability post-refining [1,25]. Specifically, the antioxidant activities for formulated blend oils ranged from 49.34% to 52.11%, with a mean value of around 51.24%, In contrast, commercial oils displayed a broader range of antioxidant activities from 40.62% to 58.68%, with some oils like single oil from rapeseed and sunflower/rapeseed blends achieving higher values. Despite this, the formulated blend oils still maintained a commendable level of antioxidant activity, underscoring their potential efficacy in various applications. For instance, formulated blends such as those involving sesame, safflower, and sunflower showed competitive antioxidant properties, demonstrating their viability as alternative options in the market.

The addition of vitamins and other antioxidants in commercial oils is a key factor that influences their superior performance in antioxidant activity results. These additives are known for their ability to inhibit oxidation processes, thus extending the oil’s usability and freshness. The study’s findings suggest that while formulated blend oils may not match the enhanced antioxidant levels of commercial oils fortified with additives, they still provide substantial antioxidant benefits, making them a valuable option for consumers seeking natural oil blends. Despite the widespread use of synthetic antioxidants to prevent lipid oxidation, there is a growing preference for natural antioxidants due to their safety profile [57].

Additionally, as mentioned above, the contrast analysis using one-way ANOVA revealed that the difference in antioxidant activity between the formulated oils and commercial oils was not significant. Once again, this indicates that the formulated blend oils are very interesting.

### 3.5. Fatty Acid Composition

Fatty acid composition, including the omega-6/omega-3 ratio, of all formulated blend oils was found to be totally different from what would be expected based on the proportion of each single seed involved in the mixture of oilseeds used for oil extraction. This indicates that the fatty acid composition of the blended oil obtained from a mixture of oilseeds cannot be predicted based only on the proportions or percentages of the individual oilseeds used, as the resulting oil becomes unique and new.

Two possible explanations could be provided, the interaction between oils and the presence and activation of enzymes. First, when oils are mixed, chemical or physical interactions between the compounds from the different seeds may occur. These interactions could lead to a redistribution of fatty acids between the oils, thereby modifying the final composition of the blend. Several studies have shown that mixing oils increased the proportions of ω-6/ω-3 fatty acids [23]. For example, it was reported that blends of fish oil and sesame seed oil contained higher levels of ω-3, ω-6, and ω-9 fatty acids than the original oils [58]. Second, certain enzymes, such as the lipases present in seeds, could be activated during the extraction process, leading to changes in fatty acid composition. These enzymes can hydrolyze triglycerides, releasing free fatty acids that may recombine differently, favoring unsaturated fatty acids such as C18:2 and C18:3 in transesterification reactions [59]. This was highlighted in a study that showed that an enzymatic cascade involving lipoxygenase, lipase, and catalase acts on the synthesis of 13-hydroperoxy linoleic acid from safflower oil in a single step [60]. Therefore, the enzymatic hydrolysis of blended oils can significantly influence the unsaturated fatty acid content, although results may vary depending on the type of oil and the specific enzymatic process used.

The obtained results showed that the fatty acid composition, including the omega-6/omega-3 ratio, of all formulated blend oils was better than that of the available commercial oils, particularly having a high percentage of oleic acid (C18:1) and a low percentage of saturated fatty acids (Figure 4). Therefore, the formulated oils have a more interesting nutritional value compared to the commercial ones. Saturated fatty acids, present in significant amounts in foods such as butter or coconut oil, can increase LDL cholesterol levels, which is harmful to cardiovascular health. In contrast, unsaturated fatty acids, such as monounsaturated and polyunsaturated fatty acids, positively affect blood lipids, notably by replacing saturated fats [23,61]. Additionally, oils rich in unsaturated fatty acids are also rich in antioxidants, which are beneficial for health. Thus, favoring oils with a low percentage of saturated fatty acids can help maintain healthy cholesterol levels and reduce the risk of cardiovascular diseases [62]. It is also noteworthy that even olive oil has a higher percentage of saturated fatty acids (15%) [63] compared to the formulated blends (10% to 13%).

The results detailed in Figure 5 provide insight into the omega-6/omega-3 ratios of our formulated blend oils compared to commercial ones. For the formulated blends, the omega-6/omega-3 ratio ranged from 2.39 to 4.02. In comparison, commercial oils, especially the blends of sunflower/soybean, sunflower/soybean/rapeseed, and sunflower/rapeseed, exhibited much higher ratios of 9.58, 8.32, and 15.46, respectively. The omega-6/omega-3 ratio is an important indicator of the nutritional quality of lipids in the diet. According to Redruello-Requejo et al. [64], it is essential to maintain a balanced ratio between these two types of fatty acids to promote health. The omega-6/omega-3 ratio should be below 4 (Boundary 1) to reduce the risk of cardiovascular diseases and around 1 to 2 (Boundary 2) to prevent obesity [62,65]. Our formulated blend oils are closest to these recommended boundaries, demonstrating their potential for offering a healthier lipid profile compared to traditional commercial oils.

A comparison of the fatty acid composition of formulated blend no 7 with that of commercial blend oils also shows that the formulated blend no 7 has superior nutritional quality (Table 3). In fact, it is richer in oleic acid and palmitoleic acid (ω7) and has the closest omega-6/omega-3 ratio to public health recommendations. Compared to olive oil, known to be a high-quality nutritional oil for consumers, the percentage of oleic acid in this blended oil (46.42%) is the closest to olive oil (55–83%), referring to the standard COI/T.15/NC No. 3/Rev. 14. November 2019.

Based on the data presented in Table 3 and Table 4, as well as Figure 1, Figure 2, Figure 3, Figure 4 and Figure 5, formulated blend no 7 (1/2 rapeseed, 1/4 sunflower, 1/8 sesame, and 1/8 safflower) demonstrates superior characteristics, including high oil yield, high total polyphenol content, high flavonoid content, high antioxidant activity, and a balanced omega-6/omega-3 ratio. Next are the formulated blends no 8 (1/8 rapeseed, 1/2 sunflower, 1/8 sesame, and 1/4 safflower) and no 9 (1/8 rapeseed, 1/8 sunflower, 1/2 sesame, and 1/4 safflower).

It is important to note that most crude oils are not directly consumable due to undesirable color and odor, necessitating a refining process. While refining improves organoleptic quality by removing undesirable secondary constituents, it also results in the loss of beneficial bioactive compounds and antioxidants. Moreover, refining can produce harmful substances such as nephrotoxic and carcinogenic mono-chloropropanediol esters and trans fatty acids [33]. Many studies have indicated that the refining process mainly impacts the total polyphenol content, reducing it by 52% to 67%, without significantly affecting the fatty acid composition [3,33,34,66,67]. Even after accounting for the potential reduction in polyphenol content due to the refining process, the formulated blends no 7, no 8, and no 9 maintain superior qualities compared to commercial oils (Table 4). Therefore, blend no 7, in particular, stands out as a nutritionally valuable alternative to olive oil for consumption, especially considering the high and increasing cost of olive oil.

## 4. Conclusions

The proposed blend formulations involve developing vegetable oil derived from a specific mixture of oilseeds in their virgin state. The study successfully demonstrates that the formulated blend oils, derived from rapeseed, sunflower, sesame, and safflower seeds, exhibited superior nutritional value compared to commercial oils, with higher levels of polyphenols, flavonoids, and a significant content of natural antioxidants. Notably, blend oil no 7 (1/2 rapeseed, 1/4 sunflower, 1/8 sesame, and 1/8 safflower) showed higher contents of oleic acid (46.4%), linoleic acid (29.7%), and linolenic acid (12.2%) and has the closest omega-6/omega-3 ratio to public health recommendations. Consequently, it offers a healthier option that potentially reduces the risk of cardiovascular diseases and prevents obesity. Additionally, the economic feasibility of producing these oils is reinforced by their substantial oil yield, which is about 36%. Moreover, even if refining were chosen, the proposed formulated blend oil (no 7) would maintain a significantly higher polyphenol content than commercial oils (12 vs. 5.5 mg GAE/100 g).

We also recommend the formulated blends no 8 (1/8 rapeseed, 1/2 sunflower, 1/8 sesame, and 1/4 safflower) and no 9 (1/8 rapeseed, 1/8 sunflower, 1/2 sesame, and 1/4 safflower), which rank just after blend oil no 7. All these combined oils are proposed to be consumed in the same way as olive oil, which is preserved without the refining process. Their consumption ensures maximum nutritional benefits and contributes significantly to overall health and wellness.

Further research and development could expand the applications of these proposed virgin blend oils, by optimizing their sensory attributes and exploring their antimicrobial properties, thereby ensuring broader market acceptance and encouraging more consumers to choose them. Additionally, it is important to consider the projection of this work into the study of enzymatic mechanisms during oil extraction, which could enable better prediction of the chemical composition at the time of extraction.

## Figures and Tables

**Figure 1 foods-13-03347-f001:**
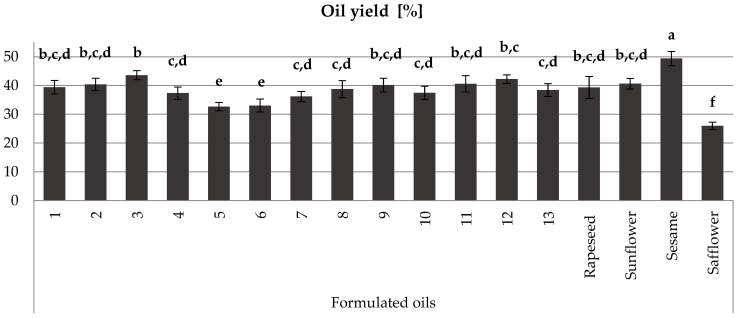
Results of the oil yield of formulated blend oils compared to oils of single seeds used in the formulation. Values are given as mean values ± standard deviation. Means values labeled with different lowercase letters differ significantly at *p* < 0.05, as determined by Duncan’s multiple range test.

**Figure 2 foods-13-03347-f002:**
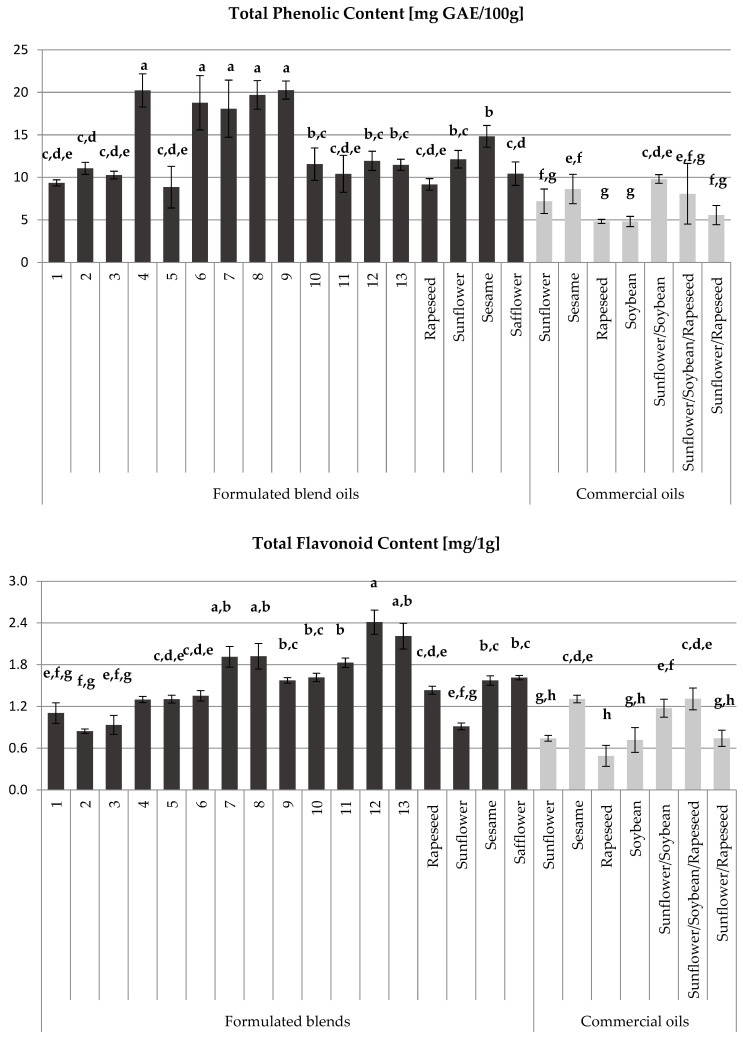
Composition of the studied oils in total polyphenol and flavonoid contents. Values are given as mean values ± standard deviation. Mean values labeled with different lowercase letters differ significantly at *p* < 0.05, as determined by Duncan’s multiple range test.

**Figure 3 foods-13-03347-f003:**
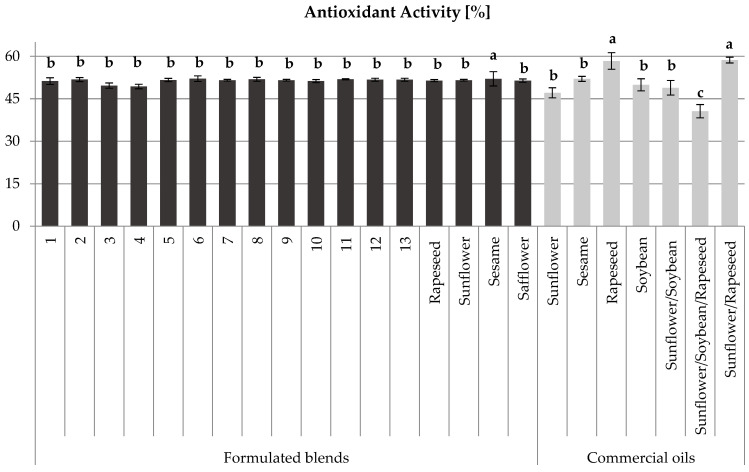
Results of the DPPH antioxidant activity analysis of formulated and commercialized oils. Values are given as mean values ± standard deviation. Mean values with different small letters are significantly different at *p* < 0.05 according to Duncan’s multiple range test.

**Figure 4 foods-13-03347-f004:**
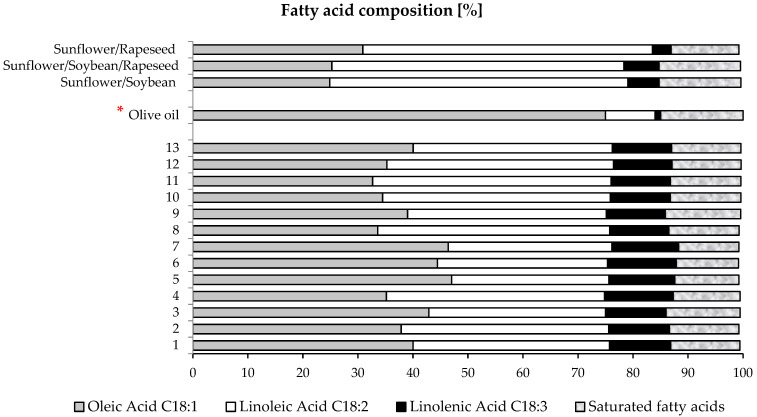
Comparison between the fatty acid composition of commercial and formulated blend oils with olive oil (* Reference: [63]).

**Figure 5 foods-13-03347-f005:**
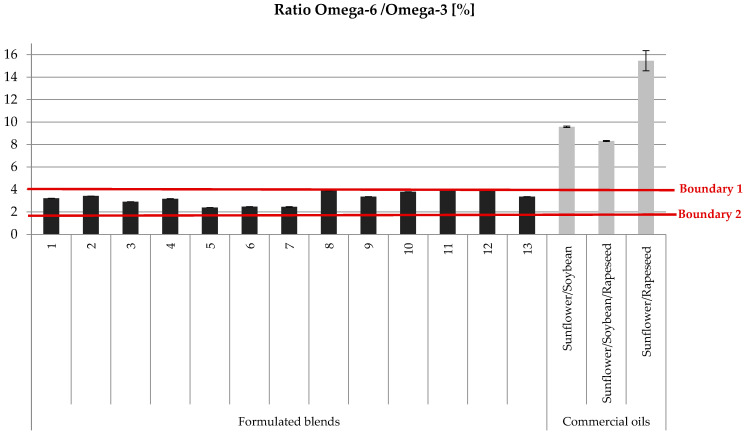
Comparison between the omega-6/omega-3 ratio of commercial and formulated oils and the recommended nutritional boundary.

**Table 1 foods-13-03347-t001:** Proportions of seeds used in the studied oils.

	Denomination	Rapeseed	Sunflower	Sesame	Safflower
Formulated oils	no 1	1/4	1/4	1/4	1/4
	no 2	1/4	1/2	1/8	1/8
	no 3	1/4	1/8	1/2	1/8
	no 4	1/4	1/8	1/8	1/2
	no 5	1/2	1/8	1/4	1/8
	no 6	1/2	1/8	1/8	1/4
	no 7	1/2	1/4	1/8	1/8
	no 8	1/8	1/2	1/8	1/4
	no 9	1/8	1/8	1/2	1/4
	no 10	1/8	1/8	1/4	1/2
	no 11	1/8	1/4	1/8	1/2
	no 12	1/8	1/2	1/4	1/8
	no 13	1/8	1/4	1/2	1/8
	Rapeseed	1	0	0	0
	Sunflower	0	1	0	0
	Sesame	0	0	1	0
	Safflower	0	0	0	1
Moroccan commercial oils	Sunflower	0	1	0	0
	Sesame	0	0	1	0
	Rapeseed	1	0	0	0
	Soybean	100% soybean			
	Sunflower/Soybean	The proportions are not specified			
	Sunflower/Soybean/Rapeseed	The proportions are not specified			
	Sunflower/Rapeseed	The proportions are not specified			

**Table 2 foods-13-03347-t002:** Results of analysis of variance for 17 formulated oils (13 blend oils and their 4 single oils) and 7 commercial oils evaluated for various quality parameters (mean square and level of difference significance).

	df	Oil Yield [%]	Acid Index [mg KOH/g]	Peroxide Index [meq O_2_/kg]	Iodine Index [g I_2_/100 g]	Polyphenol Content [mg GAE/100 g]	Flavonoid Content [mg/1 g]	Antioxidant Activity [%]	C16:1 [%]	C18:1 [%]	C18:2 [%]	C18:3 [%]	Ratio ω6/ω3
		Mean square											
Total Oils	23	54.99 ***	0.25 ***	0.72 ***	71.25 ***	42.05 ***	0.52 ***	54.41 ***	0.01 ***	119.3 ***	158.3 ***	53.3 ***	32,171.1 ***
Formulated Oils	16	75.6 ***	0.29 ***	0.96 ***	90.95 ***	53.39 ***	0.60 **	29.38 ***	0.01 ***	78.73 ***	144.2 ***	61.07 ***	7833.8 ***
Commercial Oils	6	-	0.15 ***	0.07 ***	18.72 ***	11.79 *	0.33 ***	121.2 *	0.009 ***	227.6 ***	195.8 ***	32.41 ***	97,070.4 ***
Error	48	3.694	0.020	0.022	3.360	3.896	0.149	11.53	0.003	0.014	0.020	0.002	1025.01
		Contrast value											
Formulated vs. Commercial	1	-	24.37 ***	35.22 ***	2379.1 ***	618.29 ***	5.28 ***	24.82	0.06 **	1018.4 ***	2093.5 ***	291.6 ***	70,428.8 *

df is the degree of freedom. Significant effect is indicated as *** for *p* ≤ 0.001, ** for *p* ≤ 0.01, and * for *p* ≤ 0.05. (C16:1) palmitoleic acid; (C18:1) oleic acid; (C18:2) linoleic acid; (C18:3) linolenic acid; (Ratio ω6/ω3) ratio of omega6/omega3.

**Table 3 foods-13-03347-t003:** Fatty acid composition [%] of formulated blend oil number 7 compared to Moroccan commercial oils.

Fatty Acid Name	Symbole	Formulated Blend (no 7)	Commercial Oils		
			Sunflower/Soybean	Sunflower/Soybean/ Rapeseed	Sunflower/Rapeseed
Myristic Acid	C14:0	0	0	0	0
Palmitic Acid	C16:0	6.20	9.77	10.29	7.25
Palmitoleic Acid (ω7)	C16:1	0.23	0.09	0.07	0.15
Heptadecanoic Acid	C17:0	0.04	0.03	0.07	0.06
Heptadecenoic Acid	C17:1	0.05	0.02	0.03	0.03
Stearic Acid	C18:0	3.92	4.23	4.07	3.57
Oleic Acid	C18:1	46.42	24.87	25.26	30.88
Linoleic Acid (ω6)	C18:2	29.71	54.19	53.07	52.63
Linolenic Acid (ω3)	C18:3	12.15	5.66	6.38	3.41
Arachidic Acid	C20:0	0.54	0.34	0.22	0.45
Gadoleic Acid	C20:1	0.45	0.29	0.34	0.56
Behenic Acid	C22:0	0.29	0.51	0.21	1.01
Lignoceric Acid	C24:0	0	0	0	0

**Table 4 foods-13-03347-t004:** Comparison of total polyphenol content between formulated blend oils (with and without refining) and Moroccan commercial oils.

	Category	Formulated Blends			Commercial Oils		
		(no 7)	(no 8)	(no 9)	Sunflower/Soybean	Sunflower/Soybean/Rapeseed	Sunflower/Rapeseed
Total polyphenol content [mgGAE/100 g]	Without Refining	18.08	19.70	20.27	-	-	-
	With Refining *	12.11	13.20	13.58	9.81	8.08	5.57

* Estimation of the refining process’s effect on total polyphenol composition based on the literature.

## Data Availability

The data presented in this study are available on request from the corresponding author.

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
