# Peer review of "Formulation of a High-Quality Cold-Pressed Vegetable Oil (Virgin) Based on a Blend of Four Oilseeds"

_foods, 2024, doi:10.3390/foods13213347_

Round 1
Reviewer 1 Report
Comments and Suggestions for Authors
1. Line 30: Please clarify whether virgin oil and refined oil can be defined as different extraction methods. As far as I know, refined oil is the subsequent processing process of virgin oil, and cannot be called an extraction method.
2. Line 113: sunflower has two types, edible sunflower and oil sunflower, the type of the sunflower which was used in the paper should be noted clearly.
3. Line 133-145: this paragraph was the soil and climatic conditions of four oil seeds, please explain the necessity and logic of this paragraph in this study. I don't think they are necessary in this article.
4. Line 157: Please write down the press model and experimental conditions clearly.
5. Section 2.4.1: You defined oil content as oil extracted with a mechanical extractor. The mechanical oil press can not get all the oil in the oilseed, and there will be a large amount of residual oil in the press cake, so it can not be called the oil content as you say. This is more appropriately called the oil yield.
6. Section 2.4.6: Please add the instrument and model used for absorption measurement.
7. Line 245, table 2, Please mark the units of each value clearly.
8. Line 260: Is the oil yield of different oil mixed press related to the proportion of each oil?
9. Line 335, Fig. 2: The range of the error bar is directly or even beyond the range of the data itself, which means that the fluctuation of the data is too large, is this data accurate? Repeatability is questionable.
10. The ratios of pressed oils of different oils used in your study are clear, however, the mixture ratios of commercially available products are not clear or you do not mention, then the comparability of such controlled trials is poor and insufficient to support your opinion. Is it necessary to add pressed oil in the same proportion as the commercially available blended oil for control tests?
Author Response
- Comment 1: Line 30: Please clarify whether virgin oil and refined oil can be defined as different extraction methods. As far as I know, refined oil is the subsequent processing process of virgin oil, and cannot be called an extraction method.
Response: Indeed, you are right; refining is not an extraction method but rather a processing step that occurs after extraction. The statement was edited in the revised manuscript to avoid any confusion.
- Comment 2: Line 113: Sunflower has two types, edible sunflower and oil sunflower, the type of the sunflower which was used in the paper should be noted clearly.
Response: The Lines 113 and 114 have been updated in order to be more specific as requested.
- Comment 3: Line 133-145: this paragraph was the soil and climatic conditions of four oil seeds, please explain the necessity and logic of this paragraph in this study. I don't think they are necessary in this article.
Response: Thank you for your comment. Nevertheless, several studies have shown that soil and climatic conditions can significantly influence the quality of oils. For this reason, we found it necessary to provide details on the soil and climatic conditions in our study to offer a comprehensive and relevant context for the interpretation of the results.
- Comment 4: Line 157: Please write down the press model and experimental conditions clearly.
Response: According to your suggestion, we have now specified the press model and experimental conditions in the revised manuscript.
- Comment 5: Section 2.4.1: You defined oil content as oil extracted with a mechanical extractor. The mechanical oil press can not get all the oil in the oilseed, and there will be a large amount of residual oil in the press cake, so it can not be called the oil content as you say. This is more appropriately called the oil yield.
Response: Please note that oil content was replaced with oil yield in the revised manuscript.
- Comment 6: Section 2.4.6: Please add the instrument and model used for absorption measurement.
Response: Done.
- Comment 7: Line 245, Table 2, Please mark the units of each value clearly.
Response: Done.
- Comment 8: Line 260: Is the oil yield of different oil mixed press related to the proportion of each oil?
Response: Yes, the oil yield of the mixed press is indeed related to the proportion of each oilseed in the blend as clearly shown by the findings of this study (Figure 1).
- Comment 9: Line 335, Fig. 2: The range of the error bar is directly or even beyond the range of the data itself, which means that the fluctuation of the data is too large, is this data accurate? Repeatability is questionable.
Response: Thank you very much for this relevant comment! After checking our data, we do confirm there was no error in the above subfigure 2 dedicated to total phenolic content. However, for the total flavonoid content (the below subfigure 2), we found that an error occurred in the unit of the values of the standard deviation. Instead of presenting these values in mg/1g, as for averages of total flavonoids content, they were mistakenly added in mg/100g (unit of the initial determination of this parameter). We have now corrected the standard deviation values and, thus, the error bars in Figure 2.
We ensure our data accuracy and repeatability since we performed multiple measurements. The error bars in Figure 2 represent the standard deviation calculated from three replicates conducted for each year of the 2-year study, which corresponds to six observations.
- Comment 10: The ratios of pressed oils of different oils used in your study are clear, however, the mixture ratios of commercially available products are not clear or you do not mention, then the comparability of such controlled trials is poor and insufficient to support your opinion. Is it necessary to add pressed oil in the same proportion as the commercially available blended oil for control tests?
Response: Please note that in the case of commercial oils, producers do not disclose the exact percentages of each seed used, only the types of seeds included. This is why, in Table 1, we indicated that the proportions for commercial oils are not specified, whereas we have provided the exact percentages used for the 13 formulated oil blends in our study. The objective of comparing our formulated blends with commercial oils is to highlight that we have selected seeds not commonly used by industrial crushers, as well as to demonstrate the superiority and added-value of our formulated oils over the commercial ones from a nutritional point of view. For the latter, we have analyzed all types of oils available in the Moroccan market to ensure comprehensive comparison (only 3 kinds of blended oils exist). It is also important to emphasize that our study is novel, as no similar research had been conducted on oil blends derived from a mixture of oilseeds rather than mixing extracted oils.
Reviewer 2 Report
Comments and Suggestions for Authors
The authors should address the following issues and pass through a revision for improvement. A list of questions is provided below.
1. What distinguishes cold-pressed vegetable oils from refined oils in terms of nutritional content?
2. Why was cold-pressing chosen as the method for extracting oil in this study?
3. Which blend formulation showed the highest total polyphenol and flavonoid content, and what are the specific values?
4. How does the antioxidant activity of the blend oils compare to that of commercial oils?
5. Why is the balance between omega-6 and omega-3 fatty acids important for cardiovascular health?
6. How does the lack of refining processes contribute to the overall quality of the cold-pressed vegetable oil?
7. What areas of future research are suggested to explore the sensory attributes and consumer acceptance of the formulated blend oil?
8. Why is it important to consider both nutritional and environmental factors when developing new vegetable oil formulations?
9. How could the distinctive flavors of sesame and sunflower oil contribute to the overall sensory profile of the blend?
10. What techniques could be used in future studies to measure and optimize the sensory attributes of the oil blend?
Author Response
- Question 1: What distinguishes cold-pressed vegetable oils from refined oils in terms of nutritional content?
Response: Cold-pressed oils are known for retaining more nutrients and bioactive compounds compared to refined oils. Many research studies have demonstrate that during refining, oils lose significant amounts of beneficial components, such as carotenoids (up to 98.6%), tocopherols (8.5%), phytosterols (19.5%), and squalene (45%) [Section Introduction; Line 95-97]. Cold-pressed oils, being extracted mechanically without heat or chemical treatments, preserve their natural antioxidants and Polyphenols, which contribute to their higher nutritional value [Section Results; Line 419-430].
- Question 2: Why was cold-pressing chosen as the method for extracting oil in this study?
Response: Cold-pressing was chosen for its ability to preserve the nutritional integrity of the oils. All the blended oils in our study are intended to be consumed in the same way as olive oil, which is also preserved without refining. This method ensures maximum retention of nutritional benefits, contributing significantly to overall health and wellness. Additionally, the cold-pressed blended oils may offer desirable sensory attributes, particularly with the pronounced, agreeable aromas of sesame and sunflower, which could positively influence consumer acceptance.
- Question 3: Which blend formulation showed the highest total polyphenol and flavonoid content, and what are the specific values?
Response: As demonstrated in our study (Section Results; Lines 284-310; Figure 2), several formulated blend oils were statistically classified into the same highest group. When considering both total polyphenol and flavonoid content, Blend n°7 (18.8 mg GAE/100g polyphenols and 1.91 mg/g flavonoids), Blend n°8 (19.7 mg GAE/100g polyphenols and 1.91 mg/g flavonoids), and Blend n°9 (20.2 mg GAE/100g polyphenols and 1.57 mg/g flavonoids) exhibited the highest values.
- Question 4: How does the antioxidant activity of the blend oils compare to that of commercial oils ?
Response: The antioxidant activity of the formulated blend oils ranged from 49.33% to 52.11%, demonstrating slightly lower but still competitive antioxidant activity compared to commercial oils, which ranged from 40.62% to 58.68%. The higher antioxidant activity observed in some commercial oils may be attributed to the addition of vitamins E and A, which are used to enhance shelf life and stability. Further details on this can be found in the manuscript (Section Results; Line 313-338; Figure 3).
- Question 5: Why is the balance between omega-6 and omega-3 fatty acids important for cardiovascular health ?
Response: Serval study revealed a strong association between the ratio of circulating omega-6/omega-3 PUFAs and the risk of all-cause, cancer, and cardiovascular disease mortality. The direct effect of the balance between omega-6 and omega-3 fatty acids dietary fats has been often discussed with general conclusions that this process is not simple, but rather a consequence of complex factors. Nevertheless, each group of fatty acids—SFAs, MUFAs, PUFAs and individual FAs—has a specific role in many biopathways and imbalance in their dietary intake could be result in many serious diseases. Regarding our study, we aimed to demonstrate that the formulated blend oils maintained omega-6/omega-3 ratios between 2.39 and 4.02, which aligns closely with the recommended values for reducing the risk of cardiovascular diseases (Section Results; Line 388-399; Figure 5).
- Question 6: How does the lack of refining processes contribute to the overall quality of the cold-pressed vegetable oil?
Response: The absence of refining processes allows the oils to retain their natural antioxidants and polyphenols, as refining can remove bioactive compounds that contribute to the oil’s nutritional quality and health benefits. Further details on this topic are provided in our response to Comment 1.
- Question 7: What areas of future research are suggested to explore the sensory attributes and consumer acceptance of the formulated blend oil?
Response: Future research could focus on optimizing the sensory attributes of the blend oils and assessing consumer acceptance. The distinctive flavors of the oils in the blend, particularly sesame and sunflower, offer potential for enhanced sensory appeal. We plan to conduct further sensory analysis studies to ensure consumer acceptance and market success.
- Question 8: Why is it important to consider both nutritional and environmental factors when developing new vegetable oil formulations?
Response: Considering both nutritional and environmental factors ensures that the developed oils are not only beneficial for health but also sustainable. That why in our study, the choice of oilseeds (rapeseed, sunflower, sesame, and safflower) was based on their agronomic adaptability, cost-effectiveness, and low environmental impact, particularly in light of current climate challenges. Combined with their nutritional benefits, this approach creates a product that addresses health concerns while promoting sustainability without compromise.
- Question 9: How could the distinctive flavors of sesame and sunflower oil contribute to the overall sensory profile of the blend?
Response: The sensory taste profiles of sesame and sunflower products reveal distinct characteristics influenced by their chemical compositions and processing methods. Sesame is often described as having a nutty, fruity flavor with mild bitterness, while sunflower can range from sweet to slightly rancid, depending on its quality and freshness. These contrasting flavors could combine to create a unique sensory profile that enhances the taste of the blend. This flavor complexity can appeal to consumers who seek oils with both nutritional benefits and distinctive organoleptic properties.
- Question 10: What techniques could be used in future studies to measure and optimize the sensory attributes of the oil blend?
Response: Techniques such as Quantitative Descriptive Analysis (QDA), which is an approach that uses a sensory panel testing and/or combined to Gas chromatography-olfactometry (GC-O), which integrates the separation of volatile compounds, could be used to measure and optimize the sensory attributes of the oil blend. These methods help identify key flavor compounds and assess consumer preferences, ensuring that the oil blend meets market expectations.
- Question 1: What distinguishes cold-pressed vegetable oils from refined oils in terms of nutritional content?
Response: Cold-pressed oils are known for retaining more nutrients and bioactive compounds compared to refined oils. Many research studies have demonstrate that during refining, oils lose significant amounts of beneficial components, such as carotenoids (up to 98.6%), tocopherols (8.5%), phytosterols (19.5%), and squalene (45%) [Section Introduction; Line 95-97]. Cold-pressed oils, being extracted mechanically without heat or chemical treatments, preserve their natural antioxidants and Polyphenols, which contribute to their higher nutritional value [Section Results; Line 419-430].
- Question 2: Why was cold-pressing chosen as the method for extracting oil in this study?
Response: Cold-pressing was chosen for its ability to preserve the nutritional integrity of the oils. All the blended oils in our study are intended to be consumed in the same way as olive oil, which is also preserved without refining. This method ensures maximum retention of nutritional benefits, contributing significantly to overall health and wellness. Additionally, the cold-pressed blended oils may offer desirable sensory attributes, particularly with the pronounced, agreeable aromas of sesame and sunflower, which could positively influence consumer acceptance.
- Question 3: Which blend formulation showed the highest total polyphenol and flavonoid content, and what are the specific values?
Response: As demonstrated in our study (Section Results; Lines 284-310; Figure 2), several formulated blend oils were statistically classified into the same highest group. When considering both total polyphenol and flavonoid content, Blend n°7 (18.8 mg GAE/100g polyphenols and 1.91 mg/g flavonoids), Blend n°8 (19.7 mg GAE/100g polyphenols and 1.91 mg/g flavonoids), and Blend n°9 (20.2 mg GAE/100g polyphenols and 1.57 mg/g flavonoids) exhibited the highest values.
- Question 4: How does the antioxidant activity of the blend oils compare to that of commercial oils ?
Response: The antioxidant activity of the formulated blend oils ranged from 49.33% to 52.11%, demonstrating slightly lower but still competitive antioxidant activity compared to commercial oils, which ranged from 40.62% to 58.68%. The higher antioxidant activity observed in some commercial oils may be attributed to the addition of vitamins E and A, which are used to enhance shelf life and stability. Further details on this can be found in the manuscript (Section Results; Line 313-338; Figure 3).
- Question 5: Why is the balance between omega-6 and omega-3 fatty acids important for cardiovascular health ?
Response: Serval study revealed a strong association between the ratio of circulating omega-6/omega-3 PUFAs and the risk of all-cause, cancer, and cardiovascular disease mortality. The direct effect of the balance between omega-6 and omega-3 fatty acids dietary fats has been often discussed with general conclusions that this process is not simple, but rather a consequence of complex factors. Nevertheless, each group of fatty acids—SFAs, MUFAs, PUFAs and individual FAs—has a specific role in many biopathways and imbalance in their dietary intake could be result in many serious diseases. Regarding our study, we aimed to demonstrate that the formulated blend oils maintained omega-6/omega-3 ratios between 2.39 and 4.02, which aligns closely with the recommended values for reducing the risk of cardiovascular diseases (Section Results; Line 388-399; Figure 5).
- Question 6: How does the lack of refining processes contribute to the overall quality of the cold-pressed vegetable oil?
Response: The absence of refining processes allows the oils to retain their natural antioxidants and polyphenols, as refining can remove bioactive compounds that contribute to the oil’s nutritional quality and health benefits. Further details on this topic are provided in our response to Comment 1.
- Question 7: What areas of future research are suggested to explore the sensory attributes and consumer acceptance of the formulated blend oil?
Response: Future research could focus on optimizing the sensory attributes of the blend oils and assessing consumer acceptance. The distinctive flavors of the oils in the blend, particularly sesame and sunflower, offer potential for enhanced sensory appeal. We plan to conduct further sensory analysis studies to ensure consumer acceptance and market success.
- Question 8: Why is it important to consider both nutritional and environmental factors when developing new vegetable oil formulations?
Response: Considering both nutritional and environmental factors ensures that the developed oils are not only beneficial for health but also sustainable. That why in our study, the choice of oilseeds (rapeseed, sunflower, sesame, and safflower) was based on their agronomic adaptability, cost-effectiveness, and low environmental impact, particularly in light of current climate challenges. Combined with their nutritional benefits, this approach creates a product that addresses health concerns while promoting sustainability without compromise.
- Question 9: How could the distinctive flavors of sesame and sunflower oil contribute to the overall sensory profile of the blend?
Response: The sensory taste profiles of sesame and sunflower products reveal distinct characteristics influenced by their chemical compositions and processing methods. Sesame is often described as having a nutty, fruity flavor with mild bitterness, while sunflower can range from sweet to slightly rancid, depending on its quality and freshness. These contrasting flavors could combine to create a unique sensory profile that enhances the taste of the blend. This flavor complexity can appeal to consumers who seek oils with both nutritional benefits and distinctive organoleptic properties.
- Question 10: What techniques could be used in future studies to measure and optimize the sensory attributes of the oil blend?
Response: Techniques such as Quantitative Descriptive Analysis (QDA), which is an approach that uses a sensory panel testing and/or combined to Gas chromatography-olfactometry (GC-O), which integrates the separation of volatile compounds, could be used to measure and optimize the sensory attributes of the oil blend. These methods help identify key flavor compounds and assess consumer preferences, ensuring that the oil blend meets market expectations.